# The Construction and Optimization of Ecological Security Pattern in the Harbin-Changchun Urban Agglomeration, China

**DOI:** 10.3390/ijerph16071190

**Published:** 2019-04-02

**Authors:** Rong Guo, Tong Wu, Mengran Liu, Mengshi Huang, Luigi Stendardo, Yutong Zhang

**Affiliations:** 1Key Laboratory of Cold Region Urban and Rural Human Settlement Environment Science and Technology, Ministry of Industry and Information Technology, School of Architecture, Harbin Institute of Technology, Harbin 150006, China; rongguo_hit@163.com (R.G.); mengshihuang@126.com (M.H.); 2Key Laboratory of Forest Plan Ecology, Ministry of Education, Northeast Forestry University, Harbin 150040, China; forestryliu@163.com; 3Department of Civil, Architectural and Environmental Engineering, Padova University, 35131 Padova, Italy; luigi.stendardo@unipd.it; 4Department of Agriculture and Forestry Economic Management, School of Economic Management, Northeast Forestry University, Harbin 150040, China; 13895769001@163.com

**Keywords:** ecological security pattern, ecosystem functions, least-cost path method, Harbin-Changchun urban agglomeration

## Abstract

Urban agglomerations have become a new geographical unit in China, breaking the administrative fortresses between cities, which means that the population and economic activities between cities will become more intensive in the future. Constructing and optimizing the ecological security pattern of urban agglomerations is important for promoting harmonious social-economic development and ecological protection. Using the Harbin-Changchun urban agglomeration as a case study, we have identified ecological sources based on the evaluation of ecosystem functions. Based on the resistance surface modified by nighttime light (NTL) data, the potential ecological corridors were identified using the least-cost path method, and key ecological corridors were extracted using the gravity model. By combining 15 ecological sources, 119 corridors, 3 buffer zones, and 77 ecological nodes, the ecological security pattern (ESP) was constructed. The main land-use types composed of ecological sources and corridors are forest land, cultivated land, grassland, and water areas. Some ecological sources are occupied by construction, while unused land has the potential for ecological development. The ecological corridors in the central region are distributed circularly and extend to southeast side in the form of tree branches with the Songhua River as the central axis. Finally, this study proposes an optimizing pattern with “four belts, four zones, one axis, nine corridors, ten clusters and multi-centers” to provide decision makers with spatial strategies with respect to the conflicts between urban development and ecological protection during rapid urbanization.

## 1. Introduction

Whether urban agglomerations, which constitute the main urbanization and core economic development areas in China, can develop healthily, harmoniously, and sustainably will profoundly influence the development trend of China’s social economy in the future [1,2]. In contrast to individual cities, urban agglomerations can more easily alter the ecosystem substantially in a continuous area due to decreasing or disappearing the distance between cities [3]. However, due to population growth, urban expansion, and intensive social-economic activities, the ecological land of urban agglomeration has been occupied extensively. This has been accompanied by increasing demands on natural resources, resulting in the disruption of ecological processes and thriving ecosystem functions [4,5,6]. Conversely, the destruction of ecosystems will also affect the development of the economy and society [7]. In order to coordinate the relationship between ecological protection and urbanization, researchers have begun to focus on regional ecological security studies from a spatial perspective.

The construction of an ecological security pattern (ESP) has been considered as the basic guarantee to coordinate ecosystem protection and economic development in China, the components of which were identified in a bottom-line approach to protecting priority areas and controlling urban expansion [8,9,10,11]. ESP is commonly defined as a potential spatial pattern comprising flexible and strategic elements, such as ecological source patches and corridors, with critical significance attached to safeguarding and controlling the basic ecological processes [12,13]. Under the goal of keeping ecological security in a healthy and stable state, several studies have proposed ESP as a spatial approach to integrate ecosystem functions, processes, and spatial structures [11,14]. Through an understanding of the interaction between ecological processes and landscape patterns, some specific high-priority areas can be identified as crucial components of the ESP, which correspond to the “matrix-patch-corridor” elements of landscape patterns [15]. These components play important role in providing ecosystem functions, promoting social-economic activities, and protecting animal migrations and human beings [16]. However, previous studies mainly focused on the ecological security patterns of single cities, which could not reflect the complex ecological processes of urban agglomerations without administrative boundaries’ restrictions [17]. Through the study of the interaction between spatial pattern and ecological process, ESP can not only provide basic regional protection for essential ecosystem functions, but also effectively control urban expansion [18,19].

Normally, a basic ESP can be constructed by combining the results of ecological source identification, basic resistance surface construction, and ecological corridor identification. The identification of ecological sources is the first and most fundamental step of ESP construction. Ecological sources are patches with particular ecological significance for regional ecological security, which are usually recognized by two main methods. In the first method, patches with a large area and of ecological significance are directly selected as source areas, such as nature reserves, habitats, large forestry and farmland [12,20]. Although this method is very convenient, it ignores the internal differences of the same terrain. The second method is usually based on the evaluation of ecological sensitivity, ecological suitability, ecological risk, and ecological importance. The results are reflected on spatial locations with the help of Geographic Information System (GIS) technology [21,22,23]. Currently, the most common perspective is the ecological importance evaluation based on ecological services [10]. The evaluation indicators are mainly selected according to the ecosystem functions’ characteristics of the specific study area, such as food supply, biodiversity conservation, water conservation, and soil conservation [24]. The areas with higher ecological importance are identified as ecological sources, which can be regarded as the minimum area of ecological land to meet protection of ecosystem functions.

The second step is the construction of a basic resistance surface, which can reflect the intensity of human activities and constitutes the aggregation of ecological corridors. Most studies assign a value of land cover according to expert evaluation methods or a subjective definition. However, a few studies believe that the uniform value assignment of land cover types inevitably covers up the impact of different land-use patterns and their intensity on the ecological resistance coefficient under the same land-use type [25]. It is also difficult to form a unified standard for the relative resistance values of different land types. nighttime light (NTL) data provide a good way to solve these problems, which can reflect the difference of human influence within the same land-use type [26]. Based on raster data, the intensity of human factors such as urbanization processes and population density can be synthetically characterized. Thus, the basic resistance surface can be constructed based on the value assignment of different land covers revised by NTL. 

The extraction of ecological corridors is the last step of ESP construction. Ecological corridors are optimal paths for species migration and ecological flow [27,28]. The least-cost path method of extracting ecological corridors is commonly used based on the minimum cost resistance model (MCR), which has been widely used because of its simple data structure, fast algorithms, and visualization results [10]. Originating from the study of species diffusion processes, such as mammal migration and plant seed transmission, the least-cost path method provides a spatial solution to represent human dispersal and ecological flow [29,30]. In the study of regional ecology, the minimum cumulative resistance surface obtained from the MCR model represents the minimum cumulative cost of the ecological process from sources to target points [31,32]. Considering the positions of ecological sources and the grid coefficient of the basic resistance surface, the ecological corridors can be spatially identified from the minimum cumulative resistance surface with the aid of ArcGIS [33]. Although MCR can quickly indicate the location of ecological corridors, it cannot determine the importance of corridors. Using the gravity model method, the relative importance of ecological corridors can be analyzed, which serves as an important basis for determining the priority protection order of the corridors and ecological nodes in ESP [34].

The Harbin-Changchun agglomeration is the key area for the implementation of the National New Urbanization Plan (2014–2020) in China. Its development has a stimulant effect for promoting the revitalization strategy of the old industrial base in Northeast China, the development of regional economic integration, and the transformation of resource-based cities. However, while the mode of regional economic development is rapidly transforming, the pressure on the ecological environment is increasing. According to the national Urban-Rural Construction Statistical Yearbook, during the 25 years from 1990 to 2015, urban land-use of the Harbin-Changchun agglomeration increased by 573.215 km^2^, with a growth rate of 21.05%. At present, Harbin-Changchun agglomeration is facing a conflict between land development and ecological protection. With the aim of solving this conflict, this paper takes the Harbin-Changchun agglomeration as the study area. The importance of four ecosystem functions were quantified to identify the ecological sources. Based on the results of ecological sources identification and basic resistance surface construction modified by nighttime light (NTL) data, the ecological corridors are extracted by the least-cost path method and classified based on the gravity model. Finally, an ESP is constructed, and optimization suggestions are provided for the trade-off between future land management and ecological protection.

## 2. Study Area and Data Sources

The Harbin-Changchun agglomeration (between 112°30′–116°10′ E and 29°05′~31°50′ N) is located in the northeast of China, with an area of 320,534 km^2^. As shown in Figure 1, the Harbin-Changchun agglomeration covers 10 prefecture-level cities (Harbin, Daqing, Qiqihar, Suihua, and Mudanjiang in Heilongjiang province, and Changchun, Jilin, Siping, Liaoyuan, and Songyuan in Jilin Province), one autonomous prefecture (Yanbian Korean Autonomous Prefecture in Jilin Province), 63 counties (including county-level cities), and 42 municipal districts under the jurisdiction of the 10 prefecture-level cities. The Harbin-Changchun agglomeration is located in the high-latitude monsoon climate zone, with distinct seasons, i.e., a hot and rainy summer, and a cold and dry winter. Rich geomorphological types, such as forests, cultivated land, grassland, and wetland characterize the study area.

As a region adjacent to Russia, the Korean Peninsula, and Mongolia, the Harbin-Changchun urban agglomeration plays an important role in the opening up of Northeast China to the rest of the world. It is also an important old industrial base and the largest commodity grain base in China, with a huge population and economic growth potential. By the end of 2015, the Harbin-Changchun agglomeration supported 3.4% of the national population (47 million) and contributed 3.3% of the national gross domestic product (GDP, $361.38 billion) [35]. Due to long-term human activities, the land used for such activities has experienced a rapid expansion, and an unreasonable transformation from farmland to constructed lands. For pursuing sustainability in society, the economy, and the environment, it is urgent to provide an efficient and intensive solution for promoting urbanization while protecting regional eco-environments.

Remote sensing data and statistical data were used in this study. Remote sensing data include: (1) NDVI data, and net primary productivity (NPP) data derived from NASA’s MODIS satellite products (https://modis.gsfc.nasa.gov/); (2) basic geographic data and spatial administrative boundary vector data from the National Basic Geographic Information System database (http://ngcc.sbsm.gov.cn/); (3) DEM data, downloaded from the geospatial data cloud (http://www.gscloud.cn/); (4) Land use, vegetation cover, temperature, and precipitation data are downloaded from the Resource and Environment Science Data Center of the Chinese Academy of Sciences (http://www.resdc.cn/); (5) Soil data are downloaded from the Science Data Center of Cold and Dry Areas (http://westdc.westgis.ac.cn/); (6) Defense Meteorological Satellite Program-Operational Linescan System (DMSP-OLS) nighttime light (NTL) data are from the National Geophysical Data Center of China (http://www.ngdc.noaa.gov/); (7) Spatial distribution of nature reserves and other data are derived from the China Ecosystem Assessment and Ecological Security Database (http://www.ecosystem.csdb.cn/). These data were reclassified using the nearest neighbor method, and the grid of raster data was unified to be 1000 m × 1000 m. All the statistical data come from the China Urban Statistics Yearbook 2016, Heilongjiang Statistical Yearbook 2016, and Jilin Statistical Yearbook 2016.

## 3. Methodologies

Similar to the concepts of “Green Infrastructure”, “Urban Growth Boundary”, and “Ecological Network”, an ecological security pattern can be regarded as an ensemble of the key ecological elements that are important for maintaining and controlling certain ecological processes in specific areas, such as ecological sources, surfaces, and corridors. Firstly, ecological sources are identified based on the evaluation of the importance of their ecological services. Secondly, a basic resistance surface is constructed according to different land cover types and nighttime light (NTL) data. Thirdly, ecological corridors are extracted with MCR and then classified. Finally, the ecological security pattern (ESP) is constructed based on the overlapped map of ecological sources, ecological corridors, buffer zones, and ecological nodes. The methodology framework is shown in Figure 2 and each research process is described in detail.

### 3.1. The Identification of Ecological Sources

Ecological sources refer to key natural patches that are continuously distributed in space and provide important ecosystem functions [12]. By analyzing the spatial differentiation of ecosystem functions, the importance of ecosystem functions can be evaluated, and then the regions with a higher value of importance can be selected as ecological sources. Since the study area is located in the plain area of black soil, the yield of grain crops and the forest coverage are high, and the area is rich in water resources and mineral resources. However, problems with some ecosystem functions, such as over-reclamation, biodiversity decline, water loss, and soil erosion are becoming increasingly significant. Based on the natural base and ecological environment characteristics of the ecological land in the study area, four ecosystem functions were selected to evaluate the importance value quantitatively. The four selected are food supply, biodiversity conservation, water conservation, and soil conservation. As shown in Table 1.

The importance evaluation system of ecosystem functions includes the food supply function, biodiversity conservation function, water conservation function and soil conservation function. The food supply function was calculated according to the total grain yield and the normalized differential vegetation index (NDVI) of each county administrative region. The spatial distribution was carried out by using ArcGIS spatial statistics and raster calculator module [37]. The biodiversity conservation function refers to the potential of ecosystems to provide the necessary conditions for species to survive and reproduce, which was assessed by the biodiversity module in the InVEST model (https://www. naturalcapitalproject.org/invest/) and characterized by habitat quality index [38]. The water conservation function refers to the function provided by an ecosystem’s interception of rainfall. It was quantified by the water yield module in the InVEST model [39]. The soil conservation function is evaluated by the modified soil loss equation model, and the soil conservation capacity of different regions is characterized by calculating the difference between the potential soil erosion and actual soil erosion [40].

Based on the reclassification function and the natural break-point method of ArcGIS, the calculation results of the aforementioned ecosystem functions are divided into five levels and assigned a number between 1 and 5. The greater the value, the more important the ecological function is. After the reclassification, the value of four ecosystem functions are overlapped with equal weight. As a result, the importance of the ecosystem functions was obtained and divided into five levels: extremely important, highly important, important, generally important and unimportant. In this study, the ecological patches continuity distributing in space of extremely important and highly important levels were identified and subsequently extracted as ecological sources by using the spatial analysis tool in ArcGIS.

### 3.2. The Construction of Basic Resistance Surface

Due to the influence of natural processes and human activities, distinguished characteristics exist among land-use types, which leads to different resistance to species migration and energy flow. The resistance coefficient reflects the difficulty for species migrating and energy flowing from ecological sources to sources, and mainly reflects the characteristics of different land-use types. Considering the diversity and complexity of species staying in ecological sources, the basic resistance values are not only set according to different land-use types, but also determined by net primary productivity (NPP) of vegetation, which can reflect the quality of natural environment for biological members’ survival. Based on the similar research of Yu, K.J. and other scholars, water body and construction land are assigned to minimum and maximum resistance values, which are 1 and 500 respectively [41,42,43,44]. The resistance values of other land types are obtained by normalization result of NPP data. However, the influence of different land-use patterns and intensities on the ecological resistance coefficient under the same land cover type is concealed. The NTL data coefficient of DMSP-OLS, which can synthetically characterize the intensity of human activities, is used to modify the basic resistance coefficient. The coefficient correction formula based on NTL is as follows:(1)$$R_{i}=\frac{TLI_{i}}{TLI_{a}}\times R,$$
where $R_{i}$ is the resistance coefficient modified by night light index of grid i. $TLI_{i}$ is the night light index of grid i. $TLI_{a}$ is the average night light index of land-use type corresponding to grid i, and R is the basic resistance coefficient of land-use type of grid i.

### 3.3. The Extraction and Classification of Ecological Corridors

Ecological corridors are significant channels connecting ecological sources and key paths for species migrations and ecological flows [12]. The least-cost path method can be used to determine the connections between species and energy moving between ecological sources (patches). Although studies have shown that species migration between two patches does not necessarily follow the minimum path, if species migrate along this path, they will be subject to minimal external interference or minimal resistance. The least-cost paths between sources and targets can be obtained from the minimum cumulative resistance surface, which is constructed based on the minimal cumulative resistance (MCR) model proposed by Knaappen, a Dutch ecologist [29]. The MCR model in ESP requires two input parameters: ecological security sources and ecological resistance surface. The least cost paths between the ecological security sources are tracked by the least cumulative cost through ecological resistance surface [11]. The expression of the minimum cumulative resistance model is as follows:(2)$$\mathrm{MCR}=f_{min}\sum_{j=n}^{i=m}D_{ij}\times R_{i}.$$
Here, MCR is the minimum cumulative resistance value, f is the positive correlation function, which is used to calculate the minimum resistance of any point to all sources in space. $D_{ij}$ represents the spatial distance between grid unit i and source j. $R_{i}$ is the ecological resistance coefficient of grid unit i to species movement or energy flow based on the NTL index correction.

Under the support of Cost Distance and Least Cost Path commands of ARC/Info GRID module in ArcGIS, the least-cost paths were extracted as potential ecological corridors. However, the cumulative cost of connection between different patches is different and the quality of different patches is also significantly different. Choosing which corridors to develop or protect first when optimizing the ESP is a problem. Therefore, the relative importance of potential ecological corridors and the effectiveness of links needs scientific analysis and evaluation. The gravity model method is widely used in the study of human geography at present [34]. It is usually used to measure the interaction between two homogeneous phenomena, with some common characteristics, under a certain standardized distance [45]. In the construction of an ESP, a gravity model can be used to compare the ecological importance of potential corridors in the study area and the validity of the connection, so as to determine the important ecological corridors in the study area. The gravity model is expressed as follows:(3)$$F_{mn}=\frac{P_{max}^{2}}{P_{mn}^{2}}\times \frac{\left(lnS_{m}\right)\times \left(lnS_{n}\right)}{Q_{m}\times Q_{n}},$$
where $F_{mn}$ is the interaction intensity between m and n, $P_{max}^{}$ is the maximum cumulative resistance value of all corridors in the study area, $P_{mn}^{}$ is the cumulative resistance value of potential corridors between m and n, $Q_{m}$ and $Q_{n}$ are the self-resistance values of the ecological sources m and n, and $S_{m}$ and $S_{n}$ are the areas of m and n.

Based on the matrix calculated from the gravity model, ecological corridors with interaction intensity greater than a certain threshold are extracted and regarded as important ecological corridors. The identification of the relative importance of ecological corridors, according to the interaction intensity between ecological sources, can provide a basis for determining the priority of protection of ecological corridors [46].

### 3.4. The Construction of the Ecological Security Pattern

In this paper, based on the curve of minimum cumulative resistance and area, the resistance threshold is used as the grading boundary, and the buffer zones of high, medium, and low ecological security patterns are obtained. Ecological nodes, as the key ecological strategic points for the interconnection between “sources”, are spatially reflected as the intersection points of the equal resistance lines between the ecological corridors and the minimum cumulative resistance surface, as well as the points where two or more ecological corridors intersect [47]. They are turning points for the change of material quantity and movement speed in the ecological network, and they are also the turning points for maintaining the sustainable development of ecological functions and controlling the ecology security [48]. The ESP of the Harbin-Changchun agglomeration was constructed by superimposing ecological sources, ecological corridors, buffer zones, and ecological nodes.

## 4. Results

### 4.1. Spatial Patterns of Ecosystem Functions and Ecological Sources

According to the results of our quantitative assessment of ecological services, the value of ecosystem functions was divided into five grades from 1 to 5, which means that the importance of ecosystem functions ranges from low to high. The results show that four types of ecosystem functions show spatial heterogeneity, as shown in Figure 3. The highest value of food supply services covered 14,397 km^2^, accounting for 4.5% of the total area. These are large cultivated land areas distributed in flat and open terrain. Due to high vegetation coverage and a relative absence of human activity, the highest value of biodiversity conservation services is mainly located in the Southeast. Due to abundant rainfall and forest soil with good permeability, water conservation services in the Southeast and Northeast regions are strong. Meanwhile, the storage capacity of the Songhua River and Nen River regions share the highest value of water conservation services, because of the high capacity of water bodies. The highest values of soil conservation services are mainly distributed in the low, hilly area of Southeast China, covering an area of 4775 km^2^, accounting for 1.5% of the total area. The strong soil conservation capacity in these areas is due to less human disturbance, dense vegetation, and more clay than sand.

Fifteen patches were identified as ecological sources based on the results of the importance evaluation of ecological services. The spatial distribution maps of single ecosystem function values are overlapped by equal weights, and the results of the importance of ecological services is obtained as shown in Figure 4a. Based on this result, the regions with the extremely important and highly important grades of ecosystem functions were identified as ecological sources, and their spatial distribution is shown in Figure 4b. The total area of ecological sources in the Harbin-Changchun agglomeration is 98,940.25 km^2^, accounting for 30.68% of the study area.

The ecological sources of the Harbin-Changchun urban agglomeration were mainly distributed in the hilly and mountainous areas, including the central and eastern areas of Yanbian Prefecture, the eastern areas of Jilin and Harbin, the northeastern part of Suihua, and the peripheral counties of Mudanjiang. Some of the ecological sources were distributed in the farmland areas of Changchun, Harbin, Siping, and Suihua, and the grassland areas of Daqing and Qiqihar. A small number of ecological sources were distributed in the Songhua River basin and Nen River basin of Qiqihar, Daqing, and Harbin. As the ecological bottom line to guarantee the ecological security of the Harbin-Changchun urban agglomeration, development and construction activities in these areas must be strictly prohibited. 

Among the ecological sources, forest land is the dominant land-use type, accounting for 56.24% of the total area; cultivated land, grassland, and water bodies account for 12.94%, 14.04%, and 9.8%, respectively; construction land and other land area accounted for 1.03% and 5.95%, respectively. This means that some ecological sources have been unreasonably exploited, and unused land has the potential for ecological development.

### 4.2. Spatial Patterns of Ecological Resistance and Corridors

As shown in Figure 5a, different land-use types were endowed with different cost values to construct a basic resistance surface. The resistance values of water body and construction land were 1 and 500 respectively. Based on the NPP calculation, the resistance values of other land types were revised as follows: unused land > cultivated land > grassland > woodland, and the range of values ranged from 1 to 300. The spatial distribution of the nighttime light intensity (TLI) value is shown in Figure 5b. The value of TLI is 32.95 and ranges from 0–63. The modified resistance surface of the Harbin-Changchun agglomeration was constructed based on the basic resistance surface and the modified resistance coefficient, which was the ratio of each unit’s TLI to the average TLI of the whole region. As shown in Figure 5c, the maximum value of ecological resistance was 955.99 in the Harbin-Changchun urban agglomeration in 2015. The mean resistance for Mudanjiang and Yanbian was the lowest, at below 151.35 on average. In contrast, the municipal districts of Harbin and Changchun had the highest modified resistance coefficient of above 866.80 on average. 

Based on the minimum cumulative resistance model (MCR), using the best-single module in ArcGIS, the minimum cost paths between the central points of the ecological sources were extracted. These were regarded as the potential ecological corridors between the ecological sources. As shown in Figure 5d, 119 potential ecological corridors were extracted and unevenly distributed from northwest to southeast, and from southwest to northeast, similar to the direction of Songhua River, Nen River, and the mountains. These corridors were mainly located in forest land, grassland, farmland and water areas with little human disturbance, playing a bridge role for species migration and energy flow. The total length of the potential ecological corridors is 2147.225 km, with the length of individual corridors ranging from 0.5–66.736 km. 

Large patches of ecological sources and wide ecological corridors will greatly reduce the resistance and increase the survival rate of species migrations and energy flows. The intensity of interaction between sources and targets was calculated based on the gravity model, which can characterize the effectiveness of potential ecological corridors and the importance of patches. According to the interaction matrix of 15 patches, 24 key corridors with an interaction force greater than 1000 were extracted, and the redundant corridors caused by the same patch were removed to obtain the important ecological corridors in the study area. As shown in Figure 6, the total length of the key corridors (effective corridor connections) is 429.445 km, most of which are concentrated in the central region, especially on the southeast side of Songhua River. Affected by the distribution characteristics of topography and ecological sources, the validity of ecological corridor connections in the northwest and southwest directions is low. As seen from the overall distribution, the ecological corridors in the central region distribute circularly and extend to the southeast side in the form of tree branches with the Songhua River as the central axis.

### 4.3. Construction of the Ecological Security Pattern

The ecological security pattern for the Harbin-Changchun agglomeration was constructed by overlapping ecological sources, ecological corridors, buffer zones and ecological nodes together, as shown in Figure 7. The areas of low, medium, and high-level security patterns were respectively 109,519.31 km^2^, 36,377.43 km^2^ and 20,381.68 km^2^, accounting for 33.96%, 11.28%, and 6.32% of the total area. The low-level security pattern is the rigid bottom line zone to ensure that human beings can obtain sustainable ecological services. This zone should be strictly protected in the process of urban construction, and ecological restoration and management work should be carried out to maintain the stability of the ecosystem functions and services in these areas. As for the middle-level and high-level security patterns, urbanization construction and development should be properly carried out on the basis of ensuring regional urban ecological security and arable land retention, and the development of high-level security pattern areas should be given priority. There were 77 ecological nodes identified as the most vulnerable corridor areas between ecological sources. The protection and restoration of these ecological nodes should be strengthened to ensure the smooth flow of ecological corridors. 

### 4.4. Optimization Strategy of the Ecological Security Pattern

The guideline for the development planning of Harbin-Changchun urban agglomeration, which was approved by the Chinese Government in March 2016, indicates that more economic activities will be carried out in this region [49]. This plan means that more complex material and energy flows will force new pressures onto the ecosystem. In order to ensure ecological security and sustainable development, it is urgent to optimize the ecological security pattern. In this study, forests, water bodies, and farmland are taken as ecological substrates to divide ecological function zones; river systems, transportation main lines, and natural mountains are taken as the backing to connect ecological corridors and construct ecological network system; ecological nodes are taken as strategic points to coordinate urban groups within urban agglomerations. As shown in Figure 8, the optimization model of “four belts, four districts, one axis, nine corridors and multi-centers” was put forward.

“Four belts” and “four districts” mean that the Changbai Mountains, Zhangguangcai Mountains, Xiaoxing’an Mountains, and Daxing’an Mountains are taken as ecological security protection barriers. Considering spatial differentiation characteristics of the major functions of ecological services, the ecological security pattern is divided into “four zones”, namely the water conservation zone, soil conservation zone, agri-ecological zone, and biodiversity reserve zone. Abundant broad-leaved forests widely distributed in the eastern part of the study area, providing water conservation and biodiversity maintenance services for urban development, are where forest protection and coverage should be strengthened. In this region, soil conservation capacity is also relatively strong, and vegetation coverage should be improved to control soil erosion while enhancing local economic benefits. The central part of the total area has abundant arable land resources and high food productivity, providing agricultural by-products for cities, so this is where ecological agriculture should be protected and developed. The swamp wetlands on the Song-Nen alluvial plain in the West area are widely distributed, i.e., Songnen, Sanjiang, Xianghai, and Chagan Lake, playing the role of regulating the regional ecological environment and protecting biodiversity. These wetlands should be well protected.

On the basis of ecological function zoning, the ESP optimization model of “one axis” and “nine corridors” was put forward, which means that the Songhua River is taken as the central ecological axis, protecting the key ecological corridors, as well as restoring and connecting other ecological corridors. This mode was carried out relying on the existing river system and road traffic in the total area. Priority should be given to protecting the important ecological corridors of Nen Riverthe Nen River, then the Songhua and Larin Rivers in the northwest-southeast direction, and finally the Songhua, Mudan and Tumen Rivers in the southwest-northeast direction. These river courses and coastal vegetation lands should be restored in an all-round way, and the water environment quality should be improved and protected. The aim of this is to strengthen the ecological spatial links between cities and ensure the integrity of the network of potential ecological corridors in the region. Some key artificial ecological corridors connecting Jilin, Suihua, Changchun, Qiqihar, Songyuan, Suifenhe, Hailin, and other eco-node cities (counties) along the axis of Daguang, Sui-Manch, Jingha-Heha, and Huwu Expressways should be built.

Based on the network of “one axis” and “nine corridors”, an optimized ESP mode of “ten clusters” and “multi-centers” was proposed. In this mode, Harbin and Changchun are combined as a prioritized development cluster, Daqing, Qiqihar, Suihua, Jilin, Siping, Songyuan, and Liaoyuan are regarded as a key development cluster, Yanbian and Mudanjiang are associated as an expanding development cluster. Through strengthening the connection and interchange of ecological networks, material exchange and energy flows between the internal and external ecosystems of the clusters can be enhanced. The stability of the urban-cluster-agglomeration ecosystem will form a whole and sustainable urbanization development layout. At the same time, the protection and restoration of important ecological nodes should be strengthened by local governments, through the way of promoting the transformation of traditional heavy industries to new industries with low dependence on natural resources, so as to increase government tax revenue and ensure the firm and stable pattern of ecological security.

## 5. Discussion

We identified the important components of the ESP and constructed the regional ESP of Harbin-Changchun urban agglomeration. As key components of ESP construction, ecological sources were identified based on the evaluation of ecosystem functions, ecological corridors were extracted using the least-cost path method and key ecological corridors were extracted with the gravity model. An optimizing pattern for the Harbin-Changchun urban agglomeration with “four belts, four zones, one axis, nine corridors, ten clusters and multi-centers” was proposed. 

There is growing evidence showing that construction activities usually occupy fertile farmland or forestry for construction activities, leading to the urban expansion and reduction of ecological land [5]. Our findings are consistent with several other similar EPS studies. Based on the result of ecological source identification, we found that some construction land was identified as an ecological source. This reflects the contradiction between ecosystem functions and construction activities. The study area, which formerly served as an important industry base, contained a large amount of land with a high production of food, high conversation ability of water and soil, consequently a diverse number of plots were developed into industrial land, causing the destruction of the ecological environment and a disorderly expansion of construction land. The gradual destruction of the ecological environment has seriously restricted economic development. Conversely, the backward development of economy further led to the lack of funds for governments to protect fertile farmland, soil, water, and biodiversity, thus destroying the ecological patches and corridors [7]. Therefore, the strategy of promoting the transformation of traditional heavy industries to new industries with low dependence on natural resources was proposed.

Studies on EPS construction have received increasing attention with respect to the ecological corridor extraction method, and the most frequently used one is the MCR method proposed by Knaapen for species diffusion research [29]. Adriaensen and Yu believe that this method is also used to identify the spatial location of ecological corridors which are suitable for ecological transmission and species migration based on ecological processes [32,44]. The key to extracting ecological corridors is to set resistance values in the process of resistance surface construction. The resistance surface characterizes the influence of landscape heterogeneity on the flow of ecological processes [48]. The traditional method of setting the resistance values of specific land use types to specific species neglects the difference of resistance values for other species. Considering the diversity of species and complexity of factors affecting migration resistance, we assign resistance values to different types of land use of the whole study area in two steps. In the first step, we took Yu, K.J and other scholars’ resistance values in similar studies and set the resistance values of water body and construction land to 1 and 500 respectively [41,42,43,44]. Many studies on ecological security patterns suggest that water bodies are the easiest areas and built-up areas are the most difficult areas for species migration. In the second step, we endowed resistance values of other land use types based on net primary productivity (NPP) of vegetation. According to the definition of NPP, NPP is the basic environment for survival and reproduction of biological members in the ecosystem [50]. That is to say, species survival and migration often choose areas with higher primary productivity of vegetation. We reversed NPP value to resistance value technically with the help of ArcGIS. In ArcGIS, reverse calculation and normalization of NPP data in Harbin-Changchun urban agglomeration were carried out, and the result show that resistance values of other types of land use ranging from 1 to 300. Previous studies have shown that nighttime light (NTL) data can significantly reflect the intensity of human disturbance [25,26]. Species migration and survival prefer to choose areas less affected by human activities, while avoiding areas with more human disturbance. Therefore, we used nighttime light data (NTL) to further modify values of the resistance surface. Based on the modified resistance surface, ecological corridors were selected which can both meet the requirements of species migration with suitable living environment and less human interference. 

Under the goal of keeping ecological security in a healthy state and controlling urban expansion, several studies have proposed the optimizing patterns of ESPs as a spatial approach to integrate ecosystem functions, processes, and spatial structures [11,14]. However, the existing studies on the construction and optimization of ESP mostly take the municipal administrative boundaries as the analytical boundaries, and seldom consider the spatial continuity of the geographical base and the spatial mobility of the ecosystem functions in the region [13,18,43]. Taking Harbin-Changchun urban agglomeration as the research object and providing an optimization strategy of the ESP as the strict constraints can provide guidance for urban planning decision makers to protect the ecological environment and limit the space for urban expansion.

Although this study has great significance for improving the quality of the ecological environment, controlling urban expansion, and achieving regional sustainable development, there are also some limitations. Firstly, we focus on identifying suitable locations of ecological corridors for ecological transmission and species migration based on resistance surfaces of land use types and human disturbance intensity. However, ecological corridors are not only affected by land use types, natural environment or human disturbances. Other factors also play important roles on the choice of species migration pathways, such as the distribution of target species, different abilities for species to search for resources. Especially the attraction of enemies and escape from enemies may also affect the choice of species migration pathways [51]. How to explore specific ecological corridors for specific species needs further study in the future. Secondly, based on the spatial characteristics of extracted ecological corridors, we proposed to construct and restore nine interwoven ecological corridors along the road, river and forest belts as a network to ensure the smooth migration of species and energy flow in the region. It should be noted that the width of the ecological corridor directly determines corridor function and structure. According to the existing studies on the width of ecological corridors, narrow corridors are difficult to meet the basic needs of the target species in operation, while the wide corridors need more land and increase conflicts of interest between land owners [52]. The specific width of corridors should depend on the target species and other factors (e.g. microclimate, nutrient, moisture, invasion and predation level). Due to the limitation of available data, the minimum scale of raster data used in this research is 1 km, so it is not possible to propose more detailed strategies of corridors for species conversation. We hope to make up for that in the future. Finally, based on a static perspective, this paper carries out the identification and analysis of the components of the ecological security pattern, ignoring the fact that the direction of expansion of the ecological sources will also be affected by the interaction between the surrounding cities [19]. Hence, further study on the identification of the components of the ESP based on the interaction factors of cities is required.

## 6. Conclusions

Urban agglomerations have the largest growth potential and have become the most dynamic areas in Chinese economic development. Along with the transformation of the basic regional competition units, the administrative fortresses between cities have been broken. Economic and human activities have become increasingly frequent, resulting in more threats to ecosystems. In order to solve this conflict and guide a sustainable development of urban agglomerations spatially, this paper identifies the ESP of the Harbin-Changchun urban agglomeration based on the theory of ESP, and with the help of the ArcGIS platform. The main conclusions are as follows:

(1) The importance of four ecosystem functions are quantitatively evaluated, including food supply, biodiversity conservation, water conservation, and soil conservation. The assessment shows that the area of extremely important and highly important ecological services is 98,940.25 km^2^, accounting for 30.68% of the total Harbin-Changchun urban agglomeration land area. These areas are identified as ecological sources and they are mainly distributed in the hilly and mountainous areas. This includes the central and eastern areas of Yanbian Prefecture, the eastern areas of Jilin and Harbin, the northeastern part of Suihua, and the peripheral counties of Mudanjiang. The main land-use types that compose these ecological sources are forest land, cultivated land, grassland, and water areas. The data show that some ecological sources have been unreasonably occupied by construction, while unused land has the potential for ecological development.

(2) To achieve higher accuracy, the value of basic ecological the resistance surface appointed by the same land type is modified by nighttime light data (NTL). The result shows that Mudanjiang and Yanbian have the lowest resistance, while Harbin and Changchun have the highest. The potential ecological corridors are extracted based on the MCR model. They are found to be distributed unevenly from northwest to southeast and southwest to northeast, and the key corridors are consistent with the trend of the main river systems and mountains in the territory. The total length of the 119 potential ecological corridors is 2147.225 km, and the total length of the 24 key ecological corridors is 429.445 km. Most of the ecological corridors are located on southeast side of the Songhua River. The structure of ecological corridors forms a circular network in the central area but presents a tree-like shape in peripheral area.

(3) The areas identified as low, medium, and high-level ESP each occupy a total land area of 109,519.31 km^2^, 36,377.43 km^2^, and 20,381.68 km^2^, respectively, accounting for 33.96%, 11.28% and 6.32% of the study area. The ESP optimization model of “four zones, four districts, one axis, ten groups, nine corridors and multi-centers” is proposed based on the construction of ESP. Furthermore, this optimizing pattern does not only provide an effective quantification framework to identify urban agglomeration’s ESP, but also helps urban planners and decision makers to make urban planning decisions under the guidelines of maintaining the ecological security pattern of the urban agglomeration in question.

## Figures and Tables

**Figure 1 ijerph-16-01190-f001:**
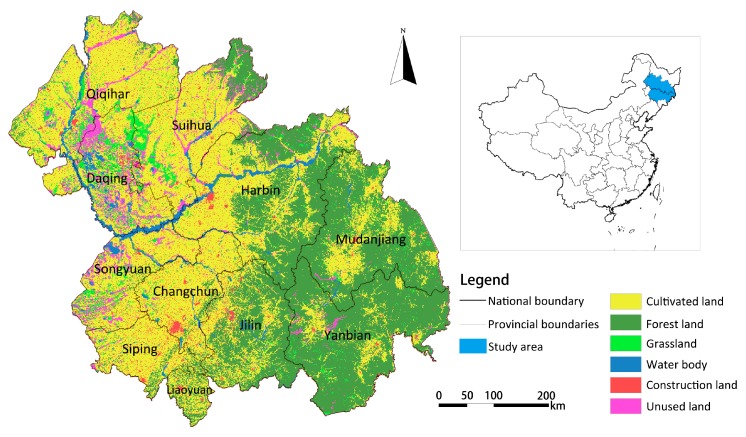
Land cover and geographical location of the Harbin-Changchun urban agglomeration.

**Figure 2 ijerph-16-01190-f002:**
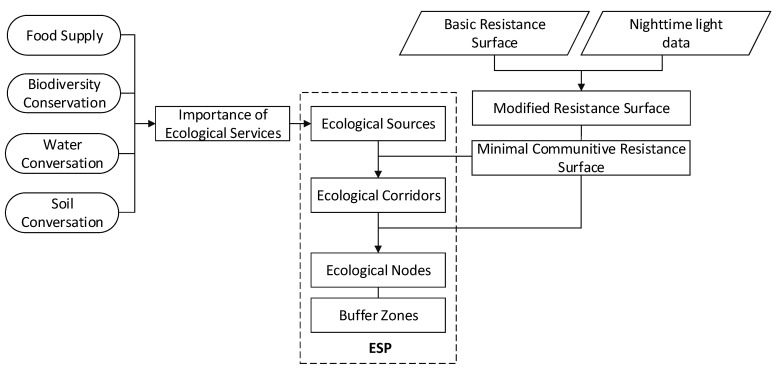
The methodology framework of the study.

**Figure 3 ijerph-16-01190-f003:**
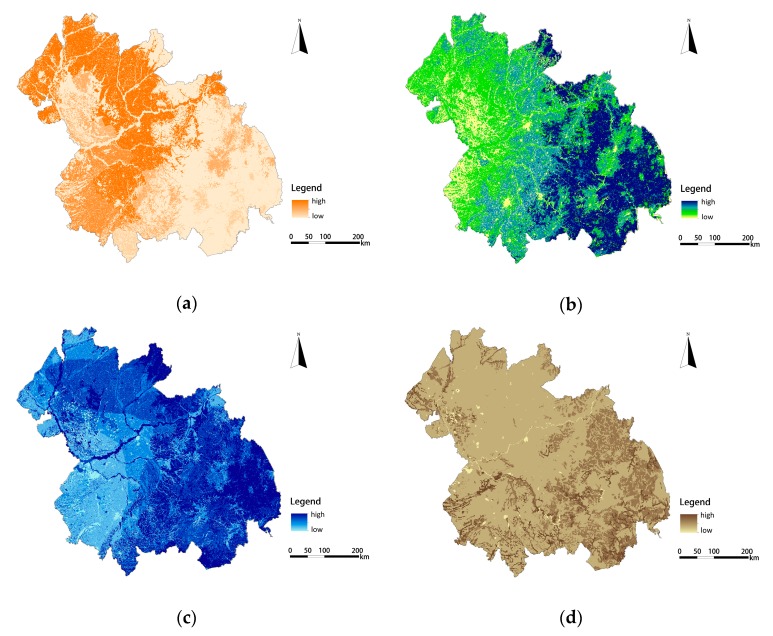
The evaluation value of ecosystem functions: (**a**) The evaluation value of food supply services; (**b**) The evaluation value of biodiversity conversation services; (**c**) The evaluation value of water conversation services; (**d**) The evaluation value of soil conversation services.

**Figure 4 ijerph-16-01190-f004:**
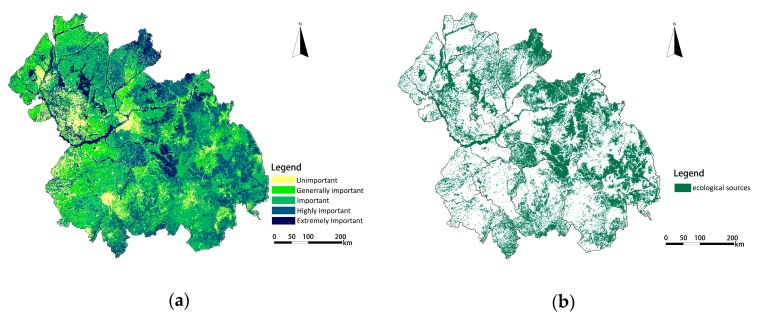
The importance grades of ecological services and their spatial distribution: (**a**) The importance grades of ecological services; (**b**) The spatial distribution of ecological sources.

**Figure 5 ijerph-16-01190-f005:**
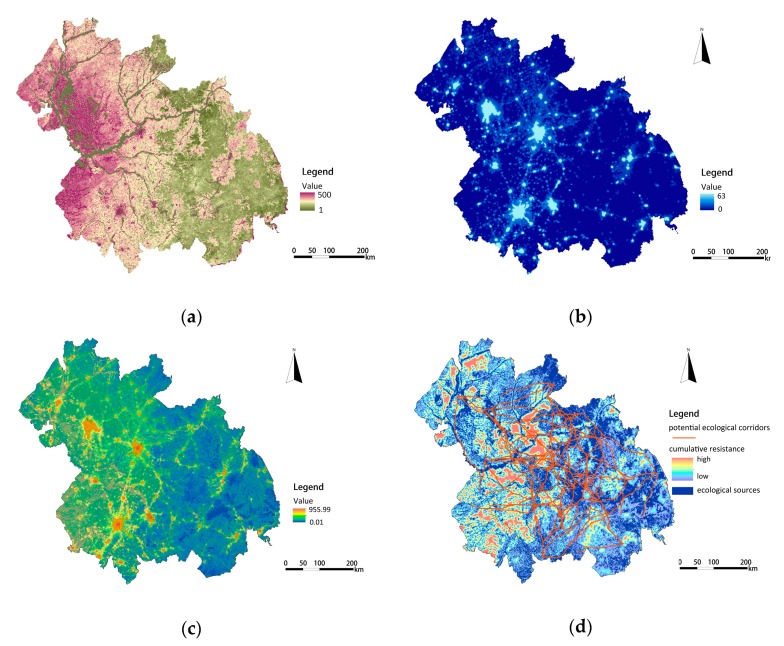
The cost value of the basic resistance surface, nighttime light intensity (TLI), and potential ecological corridors: (**a**) The cost value of the basic resistance surface; (**b**) The spatial distribution of nighttime light intensity (TLI); (**c**) The spatial distribution of the modified resistance surface; (**d**) The spatial distribution of potential ecological corridors.

**Figure 6 ijerph-16-01190-f006:**
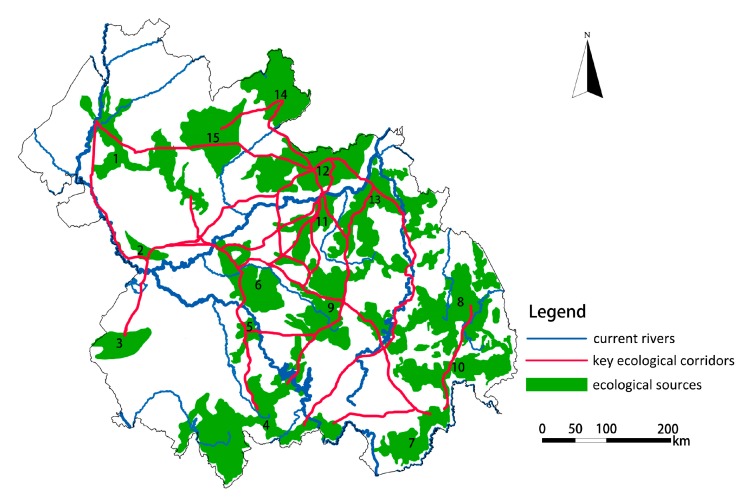
The spatial distribution of key ecological corridors.

**Figure 7 ijerph-16-01190-f007:**
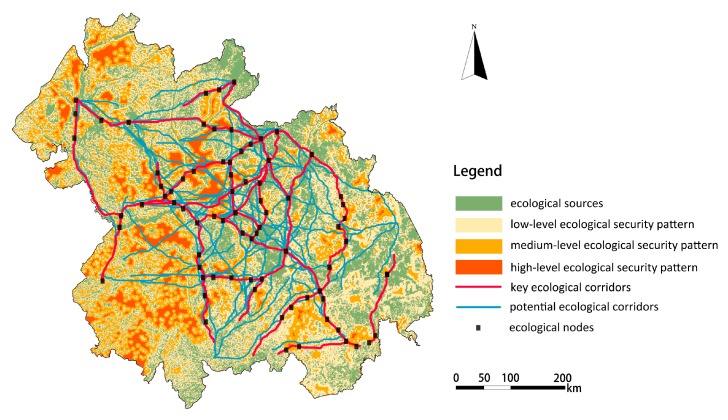
The ESP map of Harbin-Changchun urban agglomeration.

**Figure 8 ijerph-16-01190-f008:**
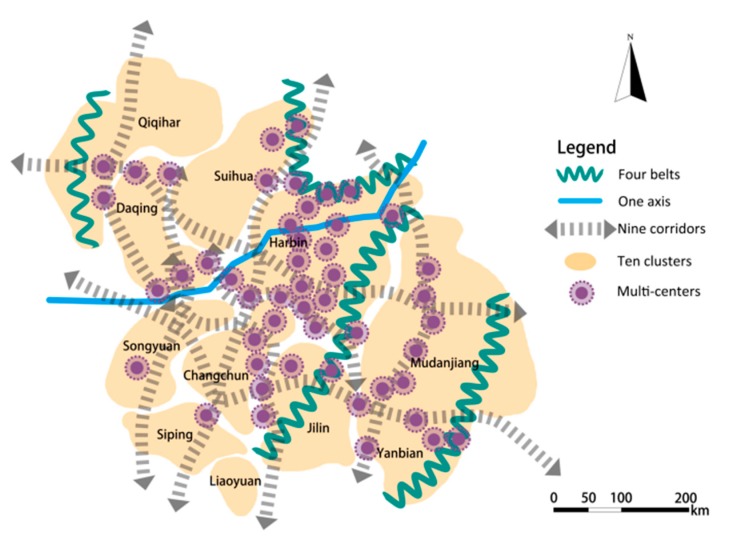
Optimized Spatial Distribution Map of ESP.

**Table 1 ijerph-16-01190-t001:** Assessment methods and required ecosystem function data.

Ecosystem Functions	Evaluation Methods	Data Needed
Food supply	$T_{x}$$=\left(NDVI_{x}/NDVI_{n}\right)\times M_{n}$ $T_{x}$ is the grain yield of grid x in county and city n, $NDVI_{x}$ is the vegetation cover index of grid x, $NDVI_{n}$ is the total vegetation cover index of county and city n, and $M_{n}$ is the total grain yield of county and city n, and is distributed using the spatial interpolation method.	NDVI data and grain production data.
biodiversity conservation	$Q_{xj}=H_{j}\left\{1-\left(\frac{D_{xj}^{z}}{D_{xj}^{z}+k^{z}}\right)\right\}$ $Q_{xj}$ is the habitat quality index of grid x in land use and land cover (habitat type) j; $D_{xj}^{}$ is the stress level of grid x in habitat type j; k is the semi-saturation constant, and half of the maximum value of $D_{xj}^{}$; $H_{j}$ is the score of the habitat type corresponding to land cover type j; the z value refers to Wu J.S.’s research, and is equal to 2.5 [36]. The $Q_{xj}$ value is directly proportional to the service capacity of biodiversity maintenance.	Land use and land cover data, and the night light data of DMSP-OLS.
Water conservation	$Y_{xj}=\left(1-\frac{AET_{xj}}{P_{x}}\right)\times P_{x}$ $Y_{xj}$ is the annual water yield, $P_{x}$ is the annual average rainfall of grid unit x, and $AET_{xj}$ is the actual annual average evapotranspiration of grid unit x on land-use type j. $Y_{xj}$ is inversely proportional to the water conservation service capacity.	Rainfall data and evapotranspiration data.
Soil conversation	A=R×K×LS−R×K×LS×C×P A is the amount of soil conservation, R×K×LS is the actual soil loss, R×K×LS×C×P is the potential soil loss, R is the rainfall and runoff factor, k is the soil erodibility factor, LS is the slope length factor, C is the vegetation cover factor, P is the factor of soil and water conservation measures. LS values are calculated based on DEM. A is proportional to the soil conservation service capacity.	Soil data and vegetation spatial distribution data.

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
