# Peer review of "The Construction and Optimization of Ecological Security Pattern in the Harbin-Changchun Urban Agglomeration, China"

_ijerph, 2019, doi:10.3390/ijerph16071190_

Round 1

Reviewer 1 Report

The Authors proposes an optimizing pattern for the Harbin-Changchun urban agglomeration with "four belts, four zones, one axis, nine corridors, ten clusters and multi-centers", providing decision makers with spatial strategies of the conflicts between urban development and ecological protection during rapid urbanization.

The paper does advance the field in a substantive way, and in my opinion is worthy of publication.

Some editorial errors are present in the manuscript:                   

1. Page 2 line 48, 49, 55, 57 and so on  – In my opinion, there should be a space between the word and the parenthesis  “… services [4-6]. “ - This should be changed throughout the article.

2. Figure 1, Figure 3-8 – In my opinion, the legibility of engravings is important and their quality should be better.

3. Page 16 line 510, 521, 526, 529, 534  and so on  –  Please change the font size, the list of literature should be improved according to the guidelines of the journal.

Author Response

Point 1: Page 2 line 48, 49, 55, 57 and so on  – In my opinion, there should be a space between the word and the parenthesis  “… services [4-6]. - This should be changed throughout the article.

Response 1: I have checked all the brackets for references numbers and added the spaces between the words and the parentheses. 

Point 2:  Figure 1, Figure 3-8 In my opinion, the legibility of engravings is important and their quality should be better.

Response 2: I am so sorry. I've replaced all the figures to high-resolution ones and canceled the image compression settings in word file. If the word file is converted to the pdf file, resolution will not be reduced.

Point 3:  Page 16 line 510, 521, 526, 529, 534  and so on  –  Please change the font size, the list of literature should be improved according to the guidelines of the journal.

Response 3: We have changed the font size, and the list of literature has been improved according to the guidelines of the journal including including doi. The revised content can be seen in lines 683-815 of the article.

In addition to the above four points, we have also made the following revisions to the article.

1. We have added 5 references (44,49,50,51,52) to support the opinions of the section Discussion.

2. In order to help readers better understand our method of obtaining ecological corridors, we added the content of method interpretation in section Methodologies, and replaced the original figure with resistance value according to land type uniformity with the figure displayed according to raster data (figure5a).The complementary content of method interpretation in section Methodologies is in lines 218-228 of the article.

3. We have moved former Discussionsection (related to ESP optimization) to Resultsection and rewrote the Discussionsection. The revised content can be found in lines 501-604 of the article.

4. We have replaced the term ecosystem serviceswith ecosystem functions. One of our considerations is that Ecosystem servicesemphasize human needs more, but we did not consider them.

Reviewer 2 Report

In relation to ecological corridors - from an ecological perspective, corridors identified  by using specific methods serve as corridors for specific species, but do not play such a role for other species or this role is difficult to estimate; this is important for conservation strategies and should be discussed for the corridors which you have designated.

Section „Discussion” doesn’t contain actual discussion of your results with the works of other authors; it is a continuation of describing of your results (related to ESP optimization); these should be moved to “Result” section, and “Discussion” should be written.

The authors use the term “ecosystem services”, but I think that ecosystem services concept is not really employed in the study. The paper should either contain more references to ecosystem services-related literature or the term "ecosystem services" should be replaced with another, e.g. "ecosystems functions".

 In the pdf file which I received  - the legends in figure 3, 4 and 5 are unreadable.

Author Response

Point 1: In relation to ecological corridors - from an ecological perspective, corridors identified  by using specific methods serve as corridors for specific species, but do not play such a role for other species or this role is difficult to estimate; this is important for conservation strategies and should be discussed for the corridors which you have designated.

Response 1: We used the method of constructing the resistance surface based on the data of net primary productivity (NPP) and nighttime light (NTL), reflecting ecological environment suitability and human interference intensity of species migration respectively. This method is different from the method of constructing resistance surfaces based on specific species. As the key of extracting ecological corridors is to set resistance values in the process of resistance surfaces construction, we added details about resistance values setting method in section Discussion, which was not written in the last manuscript. The revised content can be found in lines 525-565 of the article. In terms of the optimization strategies of ecological corridors  for species conservation, we proposed its limitation in section Discussion. The revised content can be found in lines 588-600 of the article. The revised content is as follows: 

1. Lines 525-565 of the article

Studies on the EPS construction have received increasing attention on ecological corridors extraction method and the most frequently used one is the MCR method proposed by Knaapen J.P. for species diffusion research [29]. Adriaensen F. and Yu K.J. believe that this method is also used to identify the spatial location of ecological corridors which are suitable for ecological transmission and species migration based on ecological processes [32, 44]. The key of extracting ecological corridors is to set resistance values in the process of resistance surfaces construction. The resistance surface characterizes the influence of landscape heterogeneity on the flow of ecological process [49]. The traditional method of setting resistance values of specific land use types to specific species neglects the difference of resistance values for other species. Considering diversity of species and complexity of factors affecting migration resistance, We assign resistance values to different types of land use of the whole study area in two steps. In the first step, we took Yu, K.J and other scholars' resistance values in similar studies and set the resistance values of water body and construction land to 1 and 500 respectively [41-44]. Many studies on ecological security patterns suggest that water bodies are the easiest areas and built-up areas are the most difficult areas for species migration. In the second step, we endowed resistance values of other land use types based on net primary productivity (NPP) of vegetation. According to the definition of NPP, NPP is the basic environment for survival and reproduction of biological members in the ecosystem [50]. That is to say, species survival and migration often choose areas with higher primary productivity of vegetation. We reversed NPP value to resistance value technically with the help of ArcGIS. In ArcGIS, reverse calculation and normalization of NPP data in Harbin-Changchun urban agglomeration were carried out, and the result show that resistance values of other types of land use ranging from 1 to 300. Previous studies have shown that Nighttime Light (NTL) data can significantly reflect the intensity of human disturbance [25, 26]. Species migration and survival prefer to choose areas less affected by human activities, while avoiding areas with more human disturbance. Therefore, we used Nighttime Light data (NTL) to further modify values of the resistance surface. Based on the modified resistance surface, ecological corridors were selected which can both meet the requirements of species migration with suitable living environment and less human interference.

2. Lines 588-600 of the article

Based on the spatial characteristics of extracted ecological corridors, we proposed to construct and restore nine interwoven ecological corridors along the road, river and forest belts as a network to ensure the smooth migration of species and energy flow in the region. It should be noted that the width of the ecological corridor directly determines corridor function and structure. According to the existing studies on the width of ecological corridors, narrow corridors are difficult to meet the basic needs of the target species in operation, while the wide corridors need more land and increase conflicts of interests between land owners [52]. The specific width of corridors should depend on the target species and other factors (e.g. microclimate, nutrient, moisture, invasion and predation level). Due to the limitation of available data, the minimum scale of raster data used in this research is 1 km*1 km, so it is not possible to propose more detailed optimization strategies of corridors for species conversation. We hope to make up for that in the future.

Point 2: Section Discussion doesnt contain actual discussion of your results with the works of other authors; it is a continuation of describing of your results (related to ESP optimization); these should be moved to Result section, and Discussion should be written.

Response 2: We agree with your point and we have moved former Discussion section to Result section and rewrote the Discussion section. The revised content can be found in lines 501-604 of the article. The rewrote content of Discussion section is as follows:

We identified the important components of the ESP and constructed the regional ESP of Harbin-Changchun urban agglomeration. As key components of ESP construction, ecological sources were identified based on the evaluation of ecosystem functions, ecological corridors were extracted using the least-cost path method and key ecological corridors were extracted with the gravity model. An optimizing pattern for the Harbin-Changchun urban agglomeration with "four belts, four zones, one axis, nine corridors, ten clusters and multi-centers" was proposed.

There is growing evidence showing that construction activities usually occupies fertile farmland or forestry for construction activities, leading to the urban expansion and reduction of ecological land [5]. Our findings are consistent with several other similar EPS studies. Based on the result of ecological sources identification, we found that some construction land are identified as part of ecological sources. This reflects the contradiction between ecosystem functions and construction activities. The study area, which formerly served as an important industry base, a large number of land with high production of food, high conversation ability of water and soil, high diversity of pieces were developed into industrial land, causing destruction of the ecological environment and disorderly expansion of construction land. The gradual destruction of the ecological environment has seriously restricted economic development. Conversely, the backward development of economy further lead to the lack of funds for governments to protect fertile farmland, soil, water, and biodiversity, thus destroying the ecological patches and corridors [7]. Therefore, the strategy of promoting the transformation of traditional heavy industries to new industries with low dependence on natural resources was proposed.

Studies on the EPS construction have received increasing attention on ecological corridors extraction method and the most frequently used one is the MCR method proposed by Knaapen J.P. for species diffusion research [29]. Adriaensen F. and Yu K.J. believe that this method is also used to identify the spatial location of ecological corridors which are suitable for ecological transmission and species migration based on ecological processes [32, 44]. The key of extracting ecological corridors is to set resistance values in the process of resistance surfaces construction. The resistance surface characterizes the influence of landscape heterogeneity on the flow of ecological process [49]. The traditional method of setting resistance values of specific land use types to specific species neglects the difference of resistance values for other species. Considering diversity of species and complexity of factors affecting migration resistance, We assign resistance values to different types of land use of the whole study area in two steps. In the first step, we took Yu, K.J and other scholars' resistance values in similar studies and set the resistance values of water body and construction land to 1 and 500 respectively [41-44]. Many studies on ecological security patterns suggest that water bodies are the easiest areas and built-up areas are the most difficult areas for species migration. In the second step, we endowed resistance values of other land use types based on net primary productivity (NPP) of vegetation. According to the definition of NPP, NPP is the basic environment for survival and reproduction of biological members in the ecosystem [50]. That is to say, species survival and migration often choose areas with higher primary productivity of vegetation. We reversed NPP value to resistance value technically with the help of ArcGIS. In ArcGIS, reverse calculation and normalization of NPP data in Harbin-Changchun urban agglomeration were carried out, and the result show that resistance values of other types of land use ranging from 1 to 300. Previous studies have shown that Nighttime Light (NTL) data can significantly reflect the intensity of human disturbance [25, 26]. Species migration and survival prefer to choose areas less affected by human activities, while avoiding areas with more human disturbance. Therefore, we used Nighttime Light data (NTL) to further modify values of the resistance surface. Based on the modified resistance surface, ecological corridors were selected which can both meet the requirements of species migration with suitable living environment and less human interference.

Under the goal of keeping ecological security in a healthy state and controlling urban expansion, several studies have proposed the optimizing patterns of ESPs as a spatial approach to integrate ecosystem functions, processes, and spatial structures [11, 14]. However, the existing studies on the construction and optimization of ESP mostly take the municipal administrative boundaries as the analytical boundaries, and seldom consider the spatial continuity of the geographical base and the spatial mobility of the ecosystem functions in the region [13, 18, 43]. Taking Harbin-Changchun urban agglomeration as the research object and providing optimization strategy of the ESP as the strict constraints can provide guidance for urban planning decision makers to protect ecological environment and limit the space for urban expansion.

Although this study has great significance for improving the quality of ecological environment, controlling urban expansion, and achieving regional sustainable development, there are also some limitations. Firstly, we focuses on identifying suitable locations of ecological corridors for ecological transmission and species migration based on resistance surfaces of land use types and human disturbance intensity. However, ecological corridors are not only affected by land use types, natural environment or human disturbances. Other factors also play important roles on the choice of species migration pathways, such as the distribution of target species, different abilities for species to search for resources. Especially the attraction of enemies and escape from enemies may also affect the choice of species migration pathways [51]. How to explore specific ecological corridors for specific species needs further study in the future. Secondly, Based on the spatial characteristics of extracted ecological corridors, we proposed to construct and restore nine interwoven ecological corridors along the road, river and forest belts as a network to ensure the smooth migration of species and energy flow in the region. It should be noted that the width of the ecological corridor directly determines corridor function and structure. According to the existing studies on the width of ecological corridors, narrow corridors are difficult to meet the basic needs of the target species in operation, while the wide corridors need more land and increase conflicts of interests between land owners [52]. The specific width of corridors should depend on the target species and other factors (e.g. microclimate, nutrient, moisture, invasion and predation level). Due to the limitation of available data, the minimum scale of raster data used in this research is 1 km*1 km, so it is not possible to propose more detailed corridorsoptimization strategies. We hope to make up for that in the future. Finally, based on a static perspective, this paper carries out the identification and analysis of the components of the ecological security pattern, ignoring the fact that the direction of expansion of the ecological sources will also be affected by the interaction between the surrounding cities [19]. Hence, further study on the identification of the components of the ESP based on the interaction factors of cities is required.

Point 3: The authors use the term ecosystem services, but I think that ecosystem services concept is not really employed in the study. The paper should either contain more references to ecosystem services-related literature or the term "ecosystem services" should be replaced with another, e.g. "ecosystems functions".

Response 3: We agree with your point and have replaced the term ecosystem services with ecosystem functions. One of our considerations is that Ecosystem services emphasize human needs more, but we did not consider them.

Point 4:  In the pdf file which I received  - the legends in figure 3, 4 and 5 are unreadable.

Response 4: I am so sorry. I've replaced all the figures to high-resolution ones and canceled the image compression settings in word. If the word file is converted to the pdf file, resolution will not be reduced.

In addition to the above four points, we have also made the following revisions to the article.

1. We have added 5 references (44,49,50,51,52) to support the opinions of the section Discussion.

2. In order to help readers better understand our method of obtaining ecological corridors, we added the content of method interpretation in section Methodologies, and replaced the original figure with resistance value according to land type uniformity with the figure displayed according to raster data (figure5a).The complementary content of method interpretation in section Methodologies is in lines 218-228 of the article, which is as follows:

Due to the influence of natural processes and human activities, distinguished characteristics exist among land-use types, which leads to different resistance to species migration and energy flow. The resistance coefficient reflects the difficulty for species migrating and energy flowing from ecological sources to sources, and mainly reflects the characteristics of different land-use types. Considering the diversity and complexity of species staying in ecological sources, the basic resistance values are not only set according to different land-use types, but also determined by net primary productivity (NPP) of vegetation, which can reflect the quality of natural environment for biological memberssurvival. Based on the similar research of Yu, K.J. and other scholars, water body and construction land are assigned to minimum and maximum resistance values, which are 1 and 500 respectively [41-44]. The resistance values of other land types are obtained by normalization result of NPP data. 

3. We have changed the font size, and the list of literature has been improved according to the guidelines of the journal including including doi. The revised content can be seen in lines 683-815 of the article.

Round 2

Reviewer 2 Report

-